# Cost-effectiveness of screening for chronic kidney disease using a cumulative eGFR-based statistic

**Reyhaneh Zafarnejad[1], Qiushi Chen[2], Paul M. Griffin[2]***

1 Massachusetts General Hospital, Harvard Medical School, Boston, Massachusetts, United States of America, 2 Harold and Inge Marcus Department of Industrial and Manufacturing Engineering, The Pennsylvania State University, University Park, Pennsylvania, United States of America

* pmg14@psu.edu

## Abstract

### Objectives

Routine screening for chronic kidney disease (CKD) could enable timely interventions to slow down disease progression, but currently there are no clinical guidelines for screening. We aim to evaluate the cost-effectiveness of screening for CKD using a novel analytical tool based on a cumulative sum statistic of estimated glomerular filtration rate (CUSUM_GFR).

### Methods

We developed a microsimulation model that captured CKD disease progression, major complications, patients' awareness, and treatment adherence for a nationally representative synthetic cohort of age ≥ 30 years in the United States. In addition to the status quo with no screening, we considered four CUSUM_GFR-based universal screening policies by frequency (annual or biennial) and starting age (30 or 60 years), and two targeted annual screening policies for patients with hypertension and diabetes, respectively. For each policy, we evaluated the total discounted disability-adjusted life years (DALYs) and direct health costs over a lifetime horizon and estimated the incremental cost-effectiveness ratio (ICER). We further performed one-way and probabilistic sensitivity analyses to assess the impact of parameter uncertainty.

### Results

Compared with the status quo, all the CUSUM_GFR-based screening policies were cost-effective under the willingness-to-pay (WTP) range of $50,000 –$100,000, with the estimated incremental cost-effectiveness ratios (ICERs) ranging from $15,614/DALYs averted to $54,373/DALYs averted. Universal annual screening with starting age of 30 was the non-dominated policy on the cost-effectiveness frontier under the WTP of approximately $25,000. Adding more recent treatment option of sodium–glucose cotransporter–2 (SGLT2) inhibitors to the treatment regimen was found to be cost-saving. Among the most influential

**Data Availability Statement:** Model parameters came from the literature from the National Health and Examination Survey (NHANES, https://wwwn.cdc.gov/nchs/nhanes/Default.aspx). These are defined in Table 1 of the manuscript. The source

code for the simulation model using these parameters is available at https://github.com/Rey-Zafarnejad/Cost_effectiveness_of_Screening_for_Chronic_Kidney_Disease_Using_a_Cumulative_eGFR_based_Statistic. Use of this code will allow for the replication of our findings.

**Funding:** The authors received no specific funding for this work.

**Competing interests:** The authors have declared that no competing interests exist.

model parameters, variation in the CKD progression rate, adherence, and testing cost resulted in the highest variability in model outcomes.

## Conclusions

CUSUM$_{GFR}$-based screening policies for CKD are highly cost-effective in identifying patients at risk of end stage kidney disease in early stages of CKD. Given its simple requirement of a basic blood test, the CUSUM$_{GFR}$-based screening can be easily incorporated into clinical workflow for disease monitoring and prevention.

## Introduction

Chronic kidney disease (CKD), commonly associated with diabetes or hypertension, is defined as a reduction in kidney function for at least three months of duration, measured through an estimated glomerular filtration rate (eGFR) of less than 60 mL/min per 1.73 m$^2$ or markers of kidney damage such as albuminuria [1]. Early stages of CKD are asymptomatic in most cases [2], and if unmonitored, can progress to end stage kidney disease (ESKD) defined as having eGFR below 15 mL/min per 1.73m$^2$ [3]. In the United States (US), CKD effects over one in seven adults and is responsible for approximately 16 deaths per 100,000 population, projected to become the fifth leading cause of death by 2040 [4]. In addition to disease burden, CKD has also led to a significant economic burden, accounting for 30.2% of the total Medicare expenditures in 2019 in the US [3].

Screening for CKD and implementing evidence-based interventions, especially in early stages, could result in improved clinical and societal outcomes [2, 5]. Routine screening of CKD in asymptomatic adults could not only slow down the disease progression and prevent more severe stages of CKD including ESKD, but is also beneficial in preventing other CKD-related complications, such as anemia, acute kidney conditions and nephrotoxic injuries [6]. Early identification of at-risk patients results in treatment initiation in early stages of CKD and increases the awareness of potential renal function decline, which would further reduce the rate of progression to severe stages of CKD and prevent ESKD incidence and related complications [7, 8]. While screening tests for urinary protein (micro- or macroalbuminuria) and serum creatinine (through eGFR) are often suggested for patients with mild kidney damage [9], there are currently no clinical guidelines by the US Preventive Services Task Force (USPSTF) or Community Preventive Services Task Force (CPSTF) on recommended screening policies for CKD in early stages and asymptomatic cases [10].

Despite a lack of consensus of routine testing in early stages of CKD and among asymptomatic individuals [11], screening methods that incorporate the trajectory of eGFR progression in patients with CKD have been proposed and discussed in the literature. The rationale of such a screening approach is that the renal function trajectories have been found to be more important for identifying at-risk patients than the CKD staging alone [12]. One unique advantage of eGFR-based screening is that it can be easily integrated into the clinical flow of routine outpatient or even primary care since it only requires a simple blood test with the basic metabolic panel (BPM). Given the shortage of resources in the specialty care of nephrology in the US and globally [13], implementing effective screening approaches, which require only minimum testing resources and involve primary care practitioners (PCPs), could be pivotal in helping to coordinate nephrology care for at-risk patients and save more lives.

Even though previous studies attempted to predict CKD progression using various methods [14–18], identification of at-risk patients in a cost efficient and timely manner remains challenging. This is mainly due to the significant variation in eGFR measures and resulting temporal patterns. Zafarnejad et al. [19] developed a statistical method based on the cumulative sum statistic ($CUSUM_{GFR}$), which could identify a statistically significant decrease in eGFR levels as the signal to indicate patients being at risk of ESKD and was validated using a large electronic health record (EHR) dataset. Given its desirable performance in effectively identifying at-risk patients in an earlier stage of CKD, the $CUSUM_{GFR}$ statistic offers a promising opportunity to be integrated into the screening strategy for CKD: Being easily interpretable by primary care providers, $CUSUM_{GFR}$-based screening method can leverage the existing EHR data and require limited testing resources.

In this study, our objective is to evaluate the cost-effectiveness of CKD screening policies based on the $CUSUM_{GFR}$ statistic. We developed a microsimulation to assess the total cost, health outcomes, and incremental cost-effectiveness ratio (ICER) for a variety of $CUSUM_{GFR}$-based screening policies by different frequencies, starting ages, and targeted population, compared with current practice. The results of this study could potentially provide evidence to inform effective, evidence-based guidelines for CKD screening that is currently lacking.

## Methods

We extended the CKD Health Policy Model, a previously validated and widely published microsimulation model depicting the natural history of chronic kidney disease (CKD) [20–24], by incorporating additional comorbidities to simulate disease progression, complications, and treatment outcomes for patients with CKD. A synthetic cohort with a starting age of 30 years was constructed to represent the distribution of demographics and underlying chronic conditions for the US population based on National Health and Nutrition Examination Survey (NHANES) and the United States Renal Data System (USRDS), supplemented with clinical literature [13–15]. In this study, we projected the health and cost outcomes associated with the course of disease for the lifetime horizon up to age 90, under different CKD screening policies based on the $CUSUM_{GFR}$ statistic. We further evaluated the cost-effectiveness of proposed $CUSUM_{GFR}$-based screening policies compared with the status quo and followed the Consolidated Health Economic Evaluation Reporting Standards (CHEERS) guideline for decision analytical model studies in reporting our model and results.

### Simulation model for natural history

Following a similar structure of the original CKD Health Policy Model [20–24], we simulated the progression of CKD through an annual model cycle using a synthetic cohort of individuals starting at an age of 30 years from diverse racial and ethnic groups within the US [21]. Each individual was simulated until reaching the maximum age of 90 years or death, whichever occurred earlier. The model assigned each individual the initial status of comorbid conditions including diabetes mellitus, hypertension, and proteinuria based on the prevalence in the general US population [21]. CKD stages were defined based on a combination of the presence of kidney damage markers (i.e., albuminuria in addition to eGFR $< 90$ ml/min/1.73m$^2$) or an eGFR $<60$ ml/min/1.73m$^2$ measured at least twice within a period of 3 months (with or without markers of kidney damage). The natural progression of the disease was characterized as a stochastic dynamic process with a decreasing eGFR value. The initial value of eGFR for each simulated individual was drawn from a normal distribution estimated from NHANES for individuals aged 25 to 35 [21], as all simulated individuals started at the age of 30 years. The expected value of the annual eGFR decrement depended on the patient's age and the presence

of the abovementioned comorbidities [21]. To capture the variability in eGFR values and corresponding temporal changes, the simulated eGFR decrement values were drawn from a triangular distribution [21].

The CKD Health Policy Model, upon which our study's model is based, has been previously validated, including through the use of a large electronic health record (EHR) dataset [21, 22, 24]. To further enhance the model's applicability and for structural validation purposes, we ensured that the clinical workflow embedded within the model closely aligns with existing CKD management practices. This approach includes incorporating standard diagnostic criteria, integrating relevant comorbidities into patient profiling, and ensuring consistent monitoring of kidney function throughout the simulation. Such structural validation guarantees that our simulation accurately reflects the nuances of CKD diagnosis and progression within a clinical setting. Building upon the CKD Health Policy Model, we concentrated on tailoring the model to suit our study's distinct research goals, namely introducing a novel screening method for identifying early stages of CKD and further investigating the cost-effectiveness of proposed screening strategies. Details of the natural history model parameters are provided in **S1 Table in** S1 Appendix.

We extended the original CKD Health Policy Model by incorporating major complications of CKD, patients' awareness status, and treatment adherence. Considering that anemia represents an important complication caused by CKD that affects both health and cost outcomes [24, 25], we included anemia in the disease natural history by simulating its incidence [23] and incorporated its effect on health and cost outcomes [26]. Our model explicitly captured proteinuria, diabetes, and hypertension as distinct risk factors of CKD, and considered the influence of other risk factors (such as CVD, smoking, obesity, and complications including metabolic acidosis [27], mineral bone disease [28], hyperkalemia [29]) in the aggregate form of annual medical expenditure. Moreover, we added patients' awareness of their kidney disease, since awareness may potentially influence whether patients participate in healthy behaviors (such as healthy diet and exercise) and whether they seek clinical attention [30]. The awareness rate of CKD is low in the US—less than 50% of patients with CKD stages 1 to 4 were found to be aware of their condition between 1999 to 2016 [7]. We sampled an individual's awareness status based on the awareness rate by CKD stage. In addition, we incorporated patients' adherence to prescribed treatment and the likelihood of receiving medical treatment given their medical history. Details regarding medical treatment mechanisms, patient adherence, and eligibility will be provided in the CKD treatment section.

### CUSUM$_{GFR}$-based screening

The goal of CUSUM$_{GFR}$-based screening is to detect a clinically meaningful decline in eGFR as an early indicator for the risk of progression to severe stages of CKD and ESKD. A previous study showed the validity of using a statistical measure, CUSUM$_{GFR}$, based on patients' longitudinal eGFR values—a measure that only requires a highly accessible blood test, the basic metabolic panel (BMP)—for early detection of severe CKD [19]. For an individual patient, the CUSUM$_{GFR}$ statistic was updated after each measurement and captured the historical information accumulated from all prior eGFR observations. Specifically, CUSUM$_{GFR}$ is defined as follows [19]:

$$\text{CUSUM}_{\text{GFR}_i} = \min\left[0, \left(\frac{\text{eGFR}_i - \widehat{\mu}_i}{\widehat{\sigma}}\right) + w + \text{CUSUM}_{\text{GFR}_{i-1}}\right]$$

where eGFR$_i$ indicates the eGFR measured at the $i^{\text{th}}$ observation, $\hat{\sigma}$ indicates the standard deviation of eGFR in a matched group of patients without chronic or acute kidney disease (CKD

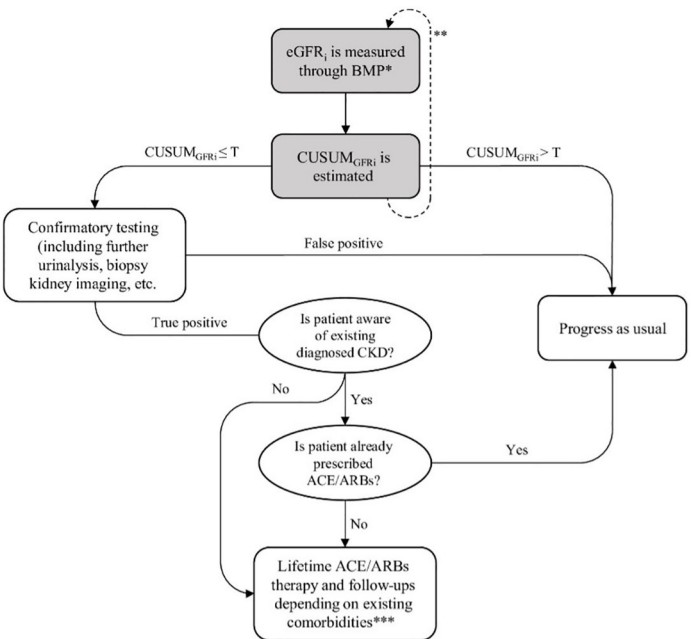

**Fig 1. Decision flowchart for CKD management following CUSUMGFR test outcomes: A patient receives treatment upon positive CUSUMGFR test result based on awareness of existing CKD, prescription history of ACE/ARBs and patient adherence.**

or AKD) diagnosis that was determined using ICD9/10 codes [19], and $\hat{\mu}_i$ is the age-adjusted mean eGFR which decreases by 0.81 mL/min/1.73 m$^2$ annually. The variable $w$ is the tuning parameter for the CUSUM$_{GFR}$ statistic. At any time, if the CUSUM$_{GFR}$ statistic fell below a certain threshold $T$, the patient was identified as being at-risk for ESKD. The optimal values for the tuning parameter $w$ and the threshold $T$ had been identified to achieve the best predictive performance from the prior study [19].

At each screening period in the simulation, individuals targeted for screening had a primary care provider visit, received an eGFR measure from a simple BMP test, and updated the corresponding CUSUM$_{GFR}$ statistic. If the CUSUM$_{GFR}$ statistic fell below the threshold, it resulted in a specialist visit and confirmatory tests (within one year) depending on the eGFR value at the signal and existing comorbidities (Fig 1). If the confirmatory test was positive, life-long monitoring and care including annual physician (generalist or nephrologist) visits and therapy would be initiated. The type of treatment and frequency of follow-up visits may vary depending on patient's existing comorbidities. For example, individuals with diabetes who were found at-risk of EKSD through CUSUM$_{GFR}$-based screening require annual nephrologist visits if their eGFR falls below 60 ml/min/1.73m$^2$. Detailed model parameters for the distributions and costs of receiving different types of confirmatory tests, treatment options, and generalist or specialist visits are provided in Table 1.

## CKD treatment

Patients confirmed to be at risk of ESKD may start to receive either Angiotensin-Converting Enzyme (ACE) inhibitors or Angiotensin II Receptor Blockers (ARBs) depending on their comorbidities and complications. Patients with either hypertension or proteinuria who have diabetes as well will receive ARBs, otherwise they will receive ACEs. If a patient does not

**Table 1. Key model parameters for the enhanced CKD simulation model.**

| Model Parameter | Parameter Value | Source |
|---|---|---|
| CUSUM$_{GFR}$ testing parameters | | [19] |
| Initial value of CUSUM$_{GFR}$ | 0 | |
| $w$ | 0.75 | |
| $T$ | -4 | |
| $\hat{\mu}_i$ | Varies by age* | |
| $\hat{\sigma}$ | 7.78 | |
| Probability of performing confirmatory testing | | [22] |
| Age < 65, proteinuria, no DM, no hypertension | 0.90 | |
| Age < 65, no proteinuria, no DM, no HTN | 0.40 | |
| Age ≥ 65, no DM, no HTN | 0.25 | |
| DM or HTN | 0.05 | |
| Awareness probability of CKD among individuals with: | | [7] |
| CKD stage 1 | 0.028 (White) 0.054 (Black) 0.009 (Hispanic) | |
| CKD stage 2 | 0.128 (White) 0.111 (Black) 0.137 (Hispanic) | |
| CKD stage 3A | 0.195 (White) 0.159 (Black) 0.146 (Hispanic) | |
| CKD stage 3B | 0.377 (White) 0.286 (Black) 0.400 (Hispanic) | |
| CKD stage 4 | 0.538 (White) 0.613 (Black) 0.833 (Hispanic) | |
| CKD stage 5 | 1.000 (White) 0.833 (Black) 0.800 (Hispanic) | |
| Treatment ineligibility | | [31] |
| Prevalence of ACE/ARBs prescription in individuals with: | | |
| Proteinuria with DM | 0.553 | |
| Proteinuria without DM | 0.337 | |
| Hypertension with DM | 0.553 | |
| Hypertension without DM | 0.337 | |
| Treatment adherence probability | 0.75 | [20] |
| Treatment hazard ratio of | | |
| ACE/ARBs on annual eGFR | 0.673 | [20, 22] |
| ACE/ARBs on mortality rate | 0.77 | [20, 22] |
| SGLT2 on annual eGFR** | 0.271 | [32] |
| SGLT2 on mortality rate | 0.74 (DM)– 0.52 (no DM) | [33] |
| Disability weights (varies by anemia diagnosis) **** | | [26] |
| No CKD | 0 | |
| CKD stage 1 and 2 | 0 | |
| CKD stage 3A | 0–0.149 | |
| CKD stage 3B | 0–0.149 | |

(*Continued*)

**Table 1.** (Continued)

| Model Parameter | Parameter Value | Source |
|---|---|---|
| CKD stage 4 | 0.104–0.237 | |
| CKD stage 5 | 0.569–0.631 | |
| ESKD, year 1 | 0.571 | |
| ESKD, year 2+ | 0.412 | |
| Health utility (annual decrements) | | [21] |
| Proteinuria | 0.01 | |
| eGFR 30–59 | 0.05 | |
| eGFR 15–29 | 0.07 | |
| eGFR <15 | 0.20 | |
| Costs | | |
| Basic Metabolic Panel (BMP) | $9.5 | [34] |
| Initial primary care visit with eGFR test | $102 | [34] |
| Confirmatory visit if first visit is positive | $80 | [34] |
| Confirmatory diagnosis costs (if eGFR < 60) | | [22, 34, 35] |
| DM | $440 | |
| HTN | $440 | |
| Proteinuria, age < 65 | $3290 | |
| Age ≥ 65 | $1111 | |
| Annual follow-up visits (if identified at-risk of ESKD) | | [22, 34, 35] |
| Specialist follow-ups (if eGFR < 60) | | |
| DM | $108 | |
| HTN or neither | $98 | |
| Generalist visits (for 3 years at least) | | |
| No DM, no HTN | $153 | |
| Annual treatment costs | | |
| ARBs | $242 | [22, 34, 35] |
| Generic ACEi | $607 | [22, 34, 35] |
| SGLT2 inhibitors (dapagliflozin or analogous) | $307 | [33] |
| General annual medical costs | Estimated via a regression model (see **S3 Table in S1 Appendix**) | [22, 36, 37] |

\* Value defined by $85.07 − 0.81\Delta t$, where $\Delta t$ is the difference in years between the age of the patient at time $i$ and point of their first eGFR measurement [19].

\*\* Value estimated by comparing the slope of eGFR decline in ACE/ARBs treatment vs combined SGLT2 and ACE/ARBs treatment groups [32]. Addition of SGLT2 inhibitors to the regimen resulted in ~40.3% reduction in eGFR slope.

\*\*\* Ranges shown reflect variation by age, sex and race/ethnicity.

\*\*\*\* CKD-caused Anemia only occurs in CKD stages 3 to 5.

Abbreviations: ACEi, angiotensin-converting enzyme inhibitor; ARB, angiotensin II receptor blockers; DM, diabetes mellitus; eGFR, estimated glomerular filtration rate; ESRD, end-stage renal disease; HTN, hypertension.; SGLT2, sodium–glucose cotransporter–2.

belong to these categories they will only be assigned to annual specialist follow-ups [22]. These treatments would slow the CKD progression through lowering the annual eGFR decline by 0.673 ml/min/1.73m$^2$ (Table 1). To prevent duplicate prescriptions, patients who were already receiving ACE inhibitors or ARBs, common medications for conditions like hypertension in CKD patients, were deemed ineligible for secondary prescription upon identification as at-risk of ESKD through the $CUSUM_{GFR}$-based screening. Whether or not a patient receives a treatment was determined by three factors: patient's awareness of their condition, patient's treatment history, and patient's adherence to the treatment regimen (Fig 1). Among those who were aware of their existing CKD condition, we assumed that patients who had already been prescribed with ACE or ARBs [31] were not eligible for additional treatment. For patients who were unaware of their condition, we assumed that they had not been treated and that in the event of a positive $CUSUM_{GFR}$ test, they became aware and thus eligible for treatment. Treatment was assumed to be initiated in only 75% of patients to account for imperfect adherence to treatment in practice [20].

## Screening policies

We defined a set of seven $CUSUM_{GFR}$-based screening policies by different combinations of screening frequencies and target population. Specifically, we considered annual and biennial screening in all individuals, which started from the age of 30 years or 60 years, respectively. We considered the option of postponing screening until the age of 60 years since age 60 and older is usually considered as an independent risk factor [38]. Furthermore, we considered two screening policies with annual $CUSUM_{GFR}$ screening targeted patients with diabetes and patients with hypertension, respectively, given that these two comorbidity conditions represent the two common risk factors of CKD. Lastly, we defined the status quo as the baseline for comparison, in which we did not perform eGFR testing as systematic screening but considered the costs and effects of clinical care for patients with CKD in the current practice.

## Health outcomes, cost, and cost-effectiveness

Our primary measure for health outcomes was disability-adjusted life years (DALYs), consisting of years of life lost (YLLs) due to premature mortality and years lived with disability (YLDs) [39]. Disability weights for each stage of CKD by the presence of the major CKD-caused complication and anemia were estimated directly from the report of Global Burden of Disease Study [26] (**S2 Table in** S1 Appendix). We included quality-adjusted life years (QALYs) as a secondary measure of health outcomes. For cost outcomes, we considered the direct medical costs related to CKD care, which included costs of testing for $CUSUM_{GFR}$, costs of follow-up treatments (if provided), and health state cost associated with managing CKD and comorbidities in the current practice (Table 1). The costs associated with screening for $CUSUM_{GFR}$, and associated follow up visits and treatment if tested positive, were derived from the literature [21, 22]. The health state costs, including the cost of inpatient, outpatient, and other medical costs associated with the care for CKD and other comorbidities, were stratified by CKD stage and estimated from a large cohort study [36] (**S3 Table in** S1 Appendix). All cost were converted to 2019 US dollars using the Consumer Price Index (CPI) [40]. We assumed the cost function holds for all the patients regardless of their awareness of the CKD condition. Both health and cost outcomes were discounted at an annual rate of 3% [41].

To assess the cost-effectiveness and the added values of CKD screening policies, we first evaluated the incremental cost-effectiveness ratio (ICER) of each $CUSUM_{GFR}$-based screening policy compared with the same comparator of the status quo (without $CUSUM_{GFR}$-based screening). We further applied the incremental analysis and characterized the cost-

effectiveness frontier among all proposed CUSUM$_{GFR}$-based screening policies to identify the most cost-effective policy under various ranges of willingness-to-pay values.

## Scenario and sensitivity analysis

To investigate the impact of emerging treatment strategies for CKD, we conducted a scenario-based analysis. This analysis was strategically focused on non-dominated screening policies due to their significant potential impact and clinical relevance, offering a clear perspective on the most promising intervention(s). Specifically, we examined the integration of sodium–glucose cotransporter–2 (SGLT2) inhibitors into the conventional treatment regimen of ACE/ARBs. SGLT2 inhibitors, initially developed for managing hyperglycemia in type 2 diabetes patients, have recently been recognized for their benefits in CKD management [32, 42]. When combined with ACE/ARBs inhibitors, these agents have shown a capacity to reduce cardiovascular mortality and heart failure events [43, 44]. Additionally, they have demonstrated a potential in slowing the progression of kidney dysfunction in patients, irrespective of their diabetic status [45, 46]. Given the limited data on SGLT2 inhibitors cost-effectiveness in CKD management and limited incorporation into standard care practices, we adopted a conservative approach in assuming that the efficacy of SGLT2 inhibitors on kidney function is comparable between patients with and without diabetes. Our analysis specifically considers the introduction of dapagliflozin or analogous SGLT2 inhibitors to the conventional ACE/ARBs treatment regimen aligning with the intervention parameters outlined in related clinical trials [33, 43, 46]. The efficacy of SGLT2 inhibitors on the rate of eGFR decline, all-cause mortality, and associated annual costs is provided in Table 1.

Additionally, we performed sensitivity analyses to evaluate the impact of parameter uncertainty on key model outcomes, which include CKD progression rate (annual decrease rate of eGFR), patient awareness, eligibility and adherence, and costs associated with CUSUM$_{GFR}$ testing and associated treatment. Given that each parameter group comprises multiple values, depending on the complications and demographics of patients as shown in **S1 Table in** S1 Appendix, we varied the values of parameters in the same group with the same percentage of changes by ±25% of their baseline values. This approach is consistent with methods used in previous studies based on the CKD Health Policy Model [22]. In addition, we conducted probabilistic sensitivity analysis (PSA) by simultaneously sampling all model parameters from defined probability distributions (**S5 Table in** S1 Appendix) [47] across 250, presenting the results in cost-effectiveness acceptability curves (CEACs) over a broad range of willingness-to-pay thresholds.

## Results

For the status quo without screening, we projected an average of 3.576 DALYs and a total cost of $123,133 per person (Table 2). All CUSUM$_{GFR}$-based screening resulted in improved health outcomes (i.e., lower DALYs or higher QALYs) with increased total costs. In comparison to the status quo, the biennial CUSUM$_{GFR}$-based screening starting at age 30 resulted in 3.448 DALYs and a total cost of $126,036 per person, resulting in 0.128 DALYs averted and an incremental cost of approximately $2,903 per person compared with the status quo. If the screening was delayed until age 60, the increments in the outcomes were projected to be lower. Increasing the frequency of the screening test from biennial to annual screening policy further increased health and cost outcomes, leading to the lowest DALY of 3.404, and the highest total cost of $126,077 when starting screening from age 30. Targeted annual screening for diabetes and hypertension population yielded lower incremental cost-effectiveness ratios than the universal screening policies starting from age 60. We also observed that the targeted screening for

**Table 2.  Base case results for cost-effectiveness of CUSUM$_{GFR}$-based screening policies.**

| Scenario | Effectiveness | | | | Lifetime Costs ($/person) | | | ICER * | |
|---|---|---|---|---|---|---|---|---|---|
| | DALYs | YLLs | YLDs | QALYs | Total costs | Screening cost | Treatment cost | $/DALY averted | $/QALY gained |
| Status quo | 3.576 | 3.404 | 0.172 | 23.605 | $123,133 | – | – | – | – |
| Universal screening | | | | | | | | | |
| Start age = 30, Biennial | 3.448 | 3.312 | 0.136 | 23.739 | $126,036 | $1,868 | $276 | $22,817 | $21,680 |
| Start age = 60, Biennial | 3.557 | 3.384 | 0.173 | 23.625 | $124,133 | $1,090 | $179 | $54,373 | $49,793 |
| Start age = 30, Annual | 3.404 | 3.243 | 0.161 | 23.794 | $126,077 | $4,406 | $960 | $17,163 | $15,614 |
| Start age = 60, Annual | 3.535 | 3.318 | 0.217 | 23.650 | $124,916 | $2,493 | $783 | $44,242 | $40,124 |
| Targeted annual screening | | | | | | | | | |
| Patients with diabetes | 3.534 | 3.362 | 0.172 | 23.645 | $124,144 | $237 | $80 | $24,054 | $25,148 |
| Patients with hypertension | 3.520 | 3.337 | 0.183 | 23.660 | $124,726 | $1,621 | $456 | $28,766 | $27,671 |

\* The ICER results were computed by comparing each policy with the status quo. See **S4 Table in** S1 Appendix for incremental analysis and Fig 2 for the cost-effectiveness frontier. Annual universal testing starting at 30 was the non-dominated policy.

*Abbreviations*: DALY: disability-adjusted life years, YLL: years of life lost due to premature mortality, YLD: years of healthy life lost due to disability, QALY: quality-adjusted life years, ICER: incremental cost-effectiveness ratio.

hypertension population was more effective than that for diabetic population. Across all CUSUM$_{GFR}$-based screening policies, the total costs were predominately attributed to the medical costs of managing CKD, comorbidities, and complications (99% of the total life-time costs), whereas the costs associated with CUSUM$_{GFR}$-based screening and the following treatments constituted the remaining 1% of the total life-time costs.

In comparison with the status quo, annual CUSUM$_{GFR}$ screening staring at age 30 resulted in the lowest ICER of $17,163/DALY-averted. By decreasing the screening frequency to biennial, the ICER increased to $22,817/DALY-averted. Delaying screening until age 60 would drastically increase the ICER to $44,242/DALY-averted for the annual screening and $54.373/DALY-averted for the biennial screening. Targeted annual screening of the diabetic population resulted in $24,054/DALY-averted compared with the status quo, while screening for the population with hypertension resulted in an ICER of $28,766/DALY-averted. Results utilizing QALYs as health measures and the ICER results based on the incremental cost per QALY-gained showed consistent findings (Table 2, S1 Fig). Considering a commonly used willingness-to-pay value ranging $50,000-$100,000/DALY-averted (or per QALY-gained) [48, 49], all of the proposed CUSUM$_{GFR}$-based screening policies were cost-effective.

The incremental analysis was performed to compare the cost-effectiveness of all screening policies. Fig 2 shows the cost-effectiveness frontier and the non-dominated screening policies. We found that universal screening starting with age 30 was the non-dominated screening policy on the frontier, and screening for the diabetic population, although a weakly dominated policy, was close to the cost-effectiveness frontier. Therefore, under the $50,000-$100,000/DALY-averted willingness-to-pay range, the universal screening starting with age 30 was deemed the most cost-effective among all CUSUM$_{GFR}$-based screening policies.

In our scenario-based analysis where SGLT2 inhibitors are added to the existing treatment regimen, annual universal screening beginning at age 30 became cost-saving compared with the status quo, despite the substantially higher annual medication costs of the new treatment regimen. When compared to the existing standard of care, the combined treatment strategy resulted in a reduction in total annual costs by approximately $516 per person. Moreover, this approach significantly enhanced health outcomes, decreasing the total DALYs and increasing the total QALYs by over 13% and 22%, respectively.

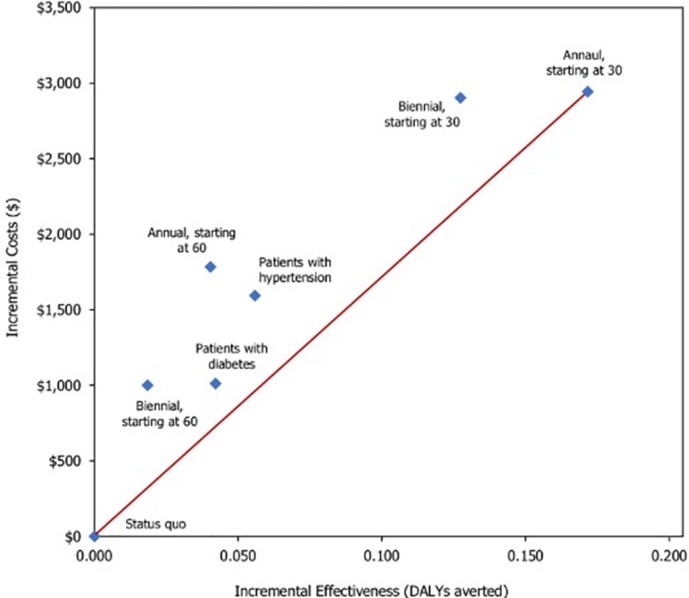

**Fig 2. Incremental cost-effectiveness analysis for CUSUMGFR-based screening for chronic kidney disease.** Annual universal screening starting at the age of 30 is the non-dominated screening policy.

Results of one-way sensitivity analysis for the non-dominated screening policy, universal screening at the age of 30 and older, and also the close second-best policy, annual screening of individuals with diabetes are presented using tornado diagrams for ICERs in Fig 3 (results for other screening policies are shown in S2 Fig). The analysis was conducted with 250 replications, and for enhanced clarity, the curves have been subjected to smoothing using a Gaussian filter. The most influential parameters across the tornado plots were CKD progression rate, adherence, and testing cost. The CEAC from the PSA (Fig 4) presents the probability that each $CUSUM_{GFR}$-based screening policies was the most cost-effective under each willingness-to-pay value. It shows that annual universal screening for CKD starting at age 30 was the most cost-effective policy with high probability for the willingness-to-pay of $24,000/DALY-averted and above.

## Discussion

The results of this study indicated that $CUSUM_{GFR}$ screening is indeed a cost-effective approach for identifying individuals at risk for ESKD through several screening strategies based on commonly used figures of $50,000 to $100,000 per DALYs-averted [48, 49]. In particular, we found that annual universal screening based on the $CUSUM_{GFR}$ method, as a conservative universal screening policy, is the non-dominated screening policy with a cost of less than $20,000/DALY averted. This is due to the comparably lower costs of performing $CUSUM_{GFR}$ through a simple BMP, and despite the impact of variation in testing and treatment costs on the cost-effectiveness outcomes, this strategy remains non-dominated over all screening policies. This emphasizes the importance of early screening for CKD not only among patients with particular risk factors but all individuals who might or might not have a known risk of CKD and further ESKD.

Given the national shortage in nephrology workforce [13], providing accurate screening approaches that can be performed by primary care practitioners (PCPs) and could play a

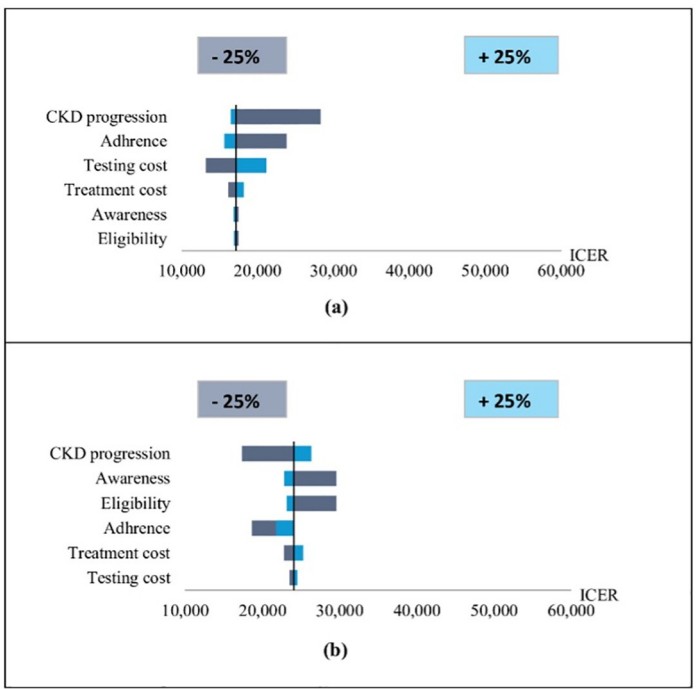

**Fig 3. One-way sensitivity analysis for the incremental cost-effectiveness ratio (ICER) of (a) annual universal screening starting at age 30 years and (b) annuals screening of individuals with diabetes, compared with the status quo without CUSUMGFR -based screening with parameter values varying within a range of 25% of the baseline value.**

critical role in helping to coordinate nephrology care for at-risk patients. Since CUSUM$_{GFR}$-based screening only requires cumulative results from BMPs in a patient's medical record, a nurse practitioner or family medicine physician would be able to interpret the results and identify at-risk patients to refer to nephrology in a timely manner. Further, universal screening would help to increase patient awareness of the condition, which has been shown to influence patient behavior including tobacco avoidance and increased physical activity [30]. Providers, and particularly PCPs, could guide identified at-risk individuals to self-manage their condition through communicating information effectively and sensitively [50], especially in more severe cases where nephrology consult necessitates the initiation of preventive medical treatment. CUSUM$_{GFR}$-based screening was found to be cost-saving when new and relatively expensive therapy options such as SGLT2 inhibitors are introduced to the treatment regimen. This could be of importance in redefining the CKD management protocols and expanding the standard of care for CKD by adding more advanced, high-efficacy therapies despite higher treatment costs. However, it is important to acknowledge that SGLT2 inhibitors, while showing promise, are still in the early stages of being integrated into CKD management and additional clinical trials and research are needed to fully establish their role. Nonetheless, our study indicates that the inclusion of SGLT2 inhibitors in standard care could be beneficial for managing early stages of CKD, suggesting a potential shift towards more proactive and effective treatment strategies in the future.

Our study has several limitations. First, as per the original Health Policy model, we assumed a linear decline in CKD progression rate by age. However, it is possible for eGFR decline to occur in a non-linear fashion [19]. When there are non-linear declines, however, they do not

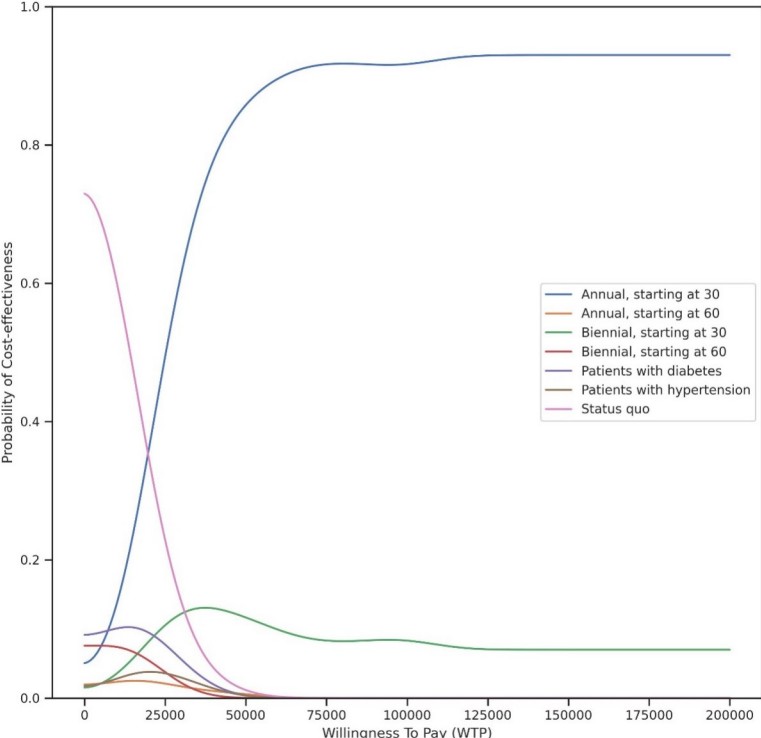

**Fig 4. Cost-effectiveness acceptability curves for all CUSUMGFR-based screening policies from probabilistic sensitivity analysis.** Universal annual screening for individuals 30 and older is the most cost-effective in comparison to the rest of the policies at the Willingness-To-Pay $\geq$ $24,000/DALYs-averted.

follow a standard pattern and so are not easily parameterized [16], which is why the linear model was selected. Second, the simulations were conducted for synthetic cohorts of individuals aged 30 and older, and comorbidities such as cardiovascular disease and stroke were only implicitly modeled through average annual medical costs and all-cause mortality rates. Third, our study parametrization relied on several data sources including the NHANES study, which despite being nationally representative, incorporates a significant proportion of self-reported information. Self-reported data may introduce biases, as they are subject to the accuracy of individual recall and personal perceptions [51]. Additionally, these datasets may not fully capture variations in awareness and comorbidities among underrepresented minority groups, potentially limiting our model's capacity to accurately reflect disparities and generalize the findings for an average population to diverse groups of different racial, ethnic, and socioeconomic backgrounds [52]. Finally, model parameters came from a variety of sources in literature, some with unknown distributions. In this case, uncertainty was modeled uniformly. Since the incremental testing and treatment costs and incremental health outcomes were relatively small, cost-effectiveness outcomes of the model were inherently sensitive to variation in parameters.

The dynamic nature of healthcare, with potential future changes in reimbursement rates, advancements in clinical methods, and evolving treatment strategies such as the emergence of SGLT2 inhibitors as new, effective drugs for CKD management, underscores the need for flexible and forward-thinking screening strategies. Moreover, there is a pressing need for practical guidelines that are fast, cost-effective, easy to interpret and implement, that do not place an

additional burden on providers and healthcare systems. Despite the availability of guidelines for screening for CKD using measurements of kidney damage (i.e., urine albumin to creatinine ratio) and function (i.e., estimated glomerular filtration rate [eGFR]) for patients with certain existing conditions [53, 54], the US Preventive Services Task Force (USPSTF) and Community Preventive Services Task Force (CPSTF) have concluded that there is not sufficient evidence for assessing the trade-off between benefits and harms of regular screening for CKD in asymptomatic adults [55]. Therefore, the results of this study could be beneficial in informing clinical practice and health policy decisions, providing insights into implementing CUSUM$_{GFR}$-based screening for early detection and management of CKD.

## Conclusions

We found annual universal screening for individuals over 30 years of age based on the CUSUM$_{GFR}$ method is cost-effective, with a cost of less than $20,000 per DALY-averted. Moreover, this screening approach is found cost-saving when novel treatment regimen like SGLT2 inhibitors are incorporated, and it stands as the non-dominated option compared to a variety of other screening strategies. Further, utilizing this screening policy could improve outcomes in persons with mild to moderate CKD and its complications, including anemia. Since the screening only requires cumulative updating of eGFR measurements from BMPs, this can help establish a framework for involving primary care practitioners in nephrology care to assist in mitigating the effect of a national shortage in nephrology workforce.

## Supporting information

**S1 Fig. Incremental cost-effectiveness analysis for CUSUMGFR-based screening for chronic kidney disease, based on QALYs.** Annual universal screening starting at the age of 30 is the non-dominated screening policy, followed closely by screening the patients with diabetes.
(TIF)

**S2 Fig. One-way sensitivity analysis results and the impact of variation in key model parameters on incremental cost-effectiveness ratio (ICER).** Parallel analysis using DALYs and QALYs resulted in analogous results.
(TIF)

**S3 Fig. Cost-effectiveness acceptability curves for all CUSUMGFR-based screening policies compared with the status quo from probabilistic sensitivity analysis.** Universal annual screening for individuals 30 and older has the highest probability of being cost-effective for the willingness-to-pay values of $24,000/DALY-averted and above. For enhanced clarity, the curves have been subjected to smoothing using a Gaussian filter.
(TIF)

**S1 Appendix.**
(DOCX)

**S1 Checklist. Human participants research checklist.**
(DOCX)

## Author Contributions

**Conceptualization:** Reyhaneh Zafarnejad, Qiushi Chen, Paul M. Griffin.

**Data curation:** Reyhaneh Zafarnejad.

**Formal analysis:** Reyhaneh Zafarnejad, Qiushi Chen, Paul M. Griffin.

**Investigation:** Qiushi Chen, Paul M. Griffin.

**Methodology:** Reyhaneh Zafarnejad, Qiushi Chen, Paul M. Griffin.

**Project administration:** Paul M. Griffin.

**Software:** Reyhaneh Zafarnejad.

**Supervision:** Qiushi Chen, Paul M. Griffin.

**Validation:** Paul M. Griffin.

**Writing – original draft:** Reyhaneh Zafarnejad, Qiushi Chen, Paul M. Griffin.

**Writing – review & editing:** Reyhaneh Zafarnejad, Qiushi Chen, Paul M. Griffin.

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
