## [Decision Letter · Decision Letter 0]

11 Dec 2023

PONE-D-23-27416Cost-effectiveness of Screening for Chronic Kidney Disease Using a Cumulative eGFR-based StatisticPLOS ONE

Dear Dr. Griffin,

Thank you for submitting your manuscript to PLOS ONE. After careful consideration, we feel that it has merit but does not fully meet PLOS ONE’s publication criteria as it currently stands. Therefore, we invite you to submit a revised version of the manuscript that addresses the points raised during the review process.

We look forward to receiving your revised manuscript.

Kind regards,

Yuri Battaglia

Academic Editor

PLOS ONE

6. We note that you have indicated that data from this study are available upon request. PLOS only allows data to be available upon request if there are legal or ethical restrictions on sharing data publicly. For more information on unacceptable data access restrictions, please see http://journals.plos.org/plosone/s/data-availability#loc-unacceptable-data-access-restrictions.

7. Please remove your figures from within your manuscript file, leaving only the individual TIFF/EPS image files, uploaded separately. These will be automatically included in the reviewers’ PDF.

Reviewers' comments:

Reviewer's Responses to Questions

**Comments to the Author**

1. Is the manuscript technically sound, and do the data support the conclusions?

Reviewer #1: Yes

Reviewer #2: Yes

2. Has the statistical analysis been performed appropriately and rigorously? 

Reviewer #1: N/A

Reviewer #2: Yes

3. Have the authors made all data underlying the findings in their manuscript fully available?

Reviewer #1: Yes

Reviewer #2: Yes

4. Is the manuscript presented in an intelligible fashion and written in standard English?

Reviewer #1: Yes

Reviewer #2: Yes

5. Review Comments to the Author

Reviewer #1: This paper is conceptually interesting because the patient's awareness is the first step to the care and the PCPs need to receive valuable tools to help their patients. It may be necessary to include the prescription of iSGLT2 like a part of the standard of care treatment for chronic kidney disease (from line 190).

Reviewer #2: The study evaluates the cost-effectiveness of screening for Chronic Kidney Disease (CKD) using a novel analytical tool, CUSUMGFR, which is based on cumulative statistics of estimated glomerular filtration rate. The research employs a microsimulation model for a U.S. cohort, considering various screening policies. Results show that all CUSUMGFR-based screening policies are cost-effective, with universal annual screening starting at age 30 being the most cost-effective. The study concludes that CUSUMGFR-based screening is highly effective in early CKD detection and can be easily integrated into clinical workflows.

The following please find comments they I suggest to improve or discuss to strengthen this study

1. Model Validation: The microsimulation model’s validity is crucial. While the model is based on a previously validated CKD Health Policy Model, additional validation specific to this study’s novel aspects, like CUSUMGFR implementation, could strengthen the findings.

2. Sensitivity Analysis: The study performs one-way and probabilistic sensitivity analyses. Expanding these analyses to include more varied scenarios could provide deeper insights into the robustness of the findings.

3. Representation of Diverse Populations: The study uses a nationally representative synthetic cohort. However, ensuring representation of diverse ethnicities and socioeconomic backgrounds could enhance the generalizability of the findings.

4. Data Sources and Limitations: While the study utilizes NHANES data, discussing the limitations of these data sources, such as potential biases or gaps, would provide a clearer understanding of the study’s scope.

5. Long-term Projection Accuracy: The study projects outcomes over a lifetime horizon. Acknowledging and addressing uncertainties in long-term projections, like changes in healthcare practices or demographics, would be beneficial.

6. Comparison with Other Screening Methods: The study focuses on CUSUMGFR-based screening. Comparing its effectiveness and cost with other potential screening methods could provide a more comprehensive view of its relative benefits.

7. Policy Implications and Practicality: While the study suggests the cost-effectiveness of CUSUMGFR-based screening, discussing the practicality of implementing such policies in real-world healthcare settings would enhance its applicability.

6. PLOS authors have the option to publish the peer review history of their article (what does this mean?). If published, this will include your full peer review and any attached files.

Reviewer #1: No

Reviewer #2: No

---

## [Author Response · Author response to Decision Letter 0]

17 Jan 2024

Responses to Reviewers’ Comments for the Manuscript “Cost-effectiveness of Screening for Chronic Kidney Disease Using a Cumulative eGFR-based Statistic” 

We are grateful for the positive feedback and constructive comments by the review team. We have carefully addressed all the questions and comments and thoroughly revised our manuscript for resubmission, which we hope has satisfactorily addressed the reviewers’ comments. In the following, we provide our point-by-point responses to each comment by the editor and reviewers.

Reviewer #1: 

This paper is conceptually interesting because the patient's awareness is the first step to the care and the PCPs need to receive valuable tools to help their patients. It may be necessary to include the prescription of iSGLT2 like a part of the standard of care treatment for chronic kidney disease (from line 190).

Response: Thank you for your insightful review. We also appreciate and agree with the reviewer’s suggestion for considering additional treatment options, such as SGLT2 inhibitors, as the cost-effectiveness of screening could eventually be driven by the subsequent treatment effectiveness and its cost implications. 

Following the reviewer’s suggestion, we added a scenario analysis to incorporate the treatment regimen of SGLT2 inhibitors in our analysis. 

(Page 13, Line 249 thorough 265, Methods): “Scenario and Sensitivity Analysis: To investigate the impact of emerging treatment strategies for CKD, we conducted a scenario-based analysis. This analysis was strategically focused on non-dominated screening policies due to their significant potential impact and clinical relevance, offering a clear perspective on the most promising intervention(s). Specifically, we examined the integration of sodium–glucose cotransporter–2 (SGLT2) inhibitors into the conventional treatment regimen of ACE/ARBs. SGLT2 inhibitors, initially developed for managing hyperglycemia in type 2 diabetes patients, have recently been recognized for their benefits in CKD management (32,38). When combined with ACE/ARBs inhibitors, these agents have shown a capacity to reduce cardiovascular mortality and heart failure events (39,40). Additionally, they have demonstrated a potential in slowing the progression of kidney dysfunction in patients, irrespective of their diabetic status (41,42). Given the limited data on SGLT2 inhibitors cost-effectiveness in CKD management and limited incorporation into standard care practices, we adopted a conservative approach in assuming that the efficacy of SGLT2 inhibitors on kidney function is comparable between patients with and without diabetes. Our analysis specifically considers the introduction of dapagliflozin or analogous SGLT2 inhibitors to the conventional ACE/ARBs treatment regimen aligning with the intervention parameters outlined in related clinical trials (33,39,42). The efficacy of SGLT2 inhibitors on the rate of eGFR decline, all-cause mortality, and associated annual costs is provided in Table 1.”

(Page 18, Line 329 thorough 333, Results): “In our scenario analysis where SGLT2 inhibitors are added to the existing treatment regimen, annual universal screening beginning at age 30 became cost-saving compared with the status quo, despite the substantially higher annual medication costs of the new treatment regimen. When compared to the existing standard of care, the combined treatment strategy resulted in a reduction in total annual costs by approximately $516 per person. Moreover, this approach significantly enhanced health outcomes, decreasing the total DALYs and increasing the total QALYs by over 13% and 22%, respectively. It is important to note, however, that while these results demonstrate the promising role of SGLT2 inhibitors in the early management of CKD, their use in this context is still emerging. Current clinical trials and ongoing research are essential to further validate these findings and assess the full potential of SGLT2 inhibitors as a standard component in CKD treatment protocols.”

(Page 21, Line 387 thorough 395, Discussion): “CUSUMGFR-based screening was found to be cost-saving when new and relatively expensive therapy options such as SGLT2 inhibitors are introduced to the treatment regimen. This could be of importance in redefining the CKD management protocols and expanding the standard of care for CKD by adding more advanced, high-efficacy therapies despite higher treatment costs. However, it is important to acknowledge that SGLT2 inhibitors, while showing promise, are still in the early stages of being integrated into CKD management and additional clinical trials and research are needed to fully establish their role. Nonetheless, our study indicates that the inclusion of SGLT2 inhibitors in standard care could be beneficial for managing early stages of CKD, suggesting a potential shift towards more proactive and effective treatment strategies in the future.”

Reviewer #2: 

The study evaluates the cost-effectiveness of screening for Chronic Kidney Disease (CKD) using a novel analytical tool, CUSUMGFR, which is based on cumulative statistics of estimated glomerular filtration rate. The research employs a microsimulation model for a U.S. cohort, considering various screening policies. Results show that all CUSUMGFR-based screening policies are cost-effective, with universal annual screening starting at age 30 being the most cost-effective. The study concludes that CUSUMGFR-based screening is highly effective in early CKD detection and can be easily integrated into clinical workflows.

The following please find comments they I suggest to improve or discuss to strengthen this study

Response: We thank the reviewer for the positive feedback on the importance of our study and constructive comments to help us further strengthen this study. We have carefully addressed each comment and provided our point-by-point responses below.

1. Model Validation: The microsimulation model’s validity is crucial. While the model is based on a previously validated CKD Health Policy Model, additional validation specific to this study’s novel aspects, like CUSUMGFR implementation, could strengthen the findings.

Response: Thank you for your detailed review. Since the main structure of the model, including the natural history of disease and management of progression, had been validated previously, in this study, we focused on adapting and extending this established model to incorporate the CUSUMGFR-based screening in clinical workflow. To address your feedback, we added more discussion in the methods section elaborating on the structural and internal validity of the clinical workflow of the model.

(Page 7, Lines 125 through 135, Methods) “The CKD Health Policy Model, upon which our study's model is based, has been previously validated, including through the use of a large electronic health record (EHR) dataset (21,22,24). To further enhance the model's applicability and for structural validation purposes, we ensured that the clinical workflow embedded within the model closely aligns with existing CKD management practices. This approach includes incorporating standard diagnostic criteria, integrating relevant comorbidities into patient profiling, and ensuring consistent monitoring of kidney function throughout the simulation. Such structural validation guarantees that our simulation accurately reflects the nuances of CKD diagnosis and progression within a clinical setting. Building upon the CKD Health Policy Model, we concentrated on tailoring the model to suit our study's distinct research goals, namely introducing a novel screening method for identifying early stages of CKD and further investigating the cost-effectiveness of proposed screening strategies. Details of the natural history model parameters are provided in Table S1.”

2. Sensitivity Analysis: The study performs one-way and probabilistic sensitivity analyses. Expanding these analyses to include more varied scenarios could provide deeper insights into the robustness of the findings.

Response: Thank you for your comment. We conducted one-way sensitivity analysis to systematically evaluate the impact of each group of parameters on the cost-effectiveness outcomes and presented the tornado diagram in Figure 3. Inspired by the comment from Reviewer 1, we included an additional scenario in which the SGLT2 inhibitors are added to the conventional ACE/ARB treatment regimen in the non-dominated screening scenario. We found that adding SGLT2 to the conventional CKD treatment regimen is cost-saving in this additional scenario. 

We have revised the main text accordingly to incorporate the descriptions and the results of the new scenario:

(Page 13, Line 249 thorough 265, Methods): “Scenario and Sensitivity Analysis: To investigate the impact of emerging treatment strategies for CKD, we conducted a scenario-based analysis. This analysis was strategically focused on non-dominated screening policies due to their significant potential impact and clinical relevance, offering a clear perspective on the most promising intervention(s). Specifically, we examined the integration of sodium–glucose cotransporter–2 (SGLT2) inhibitors into the conventional treatment regimen of ACE/ARBs. SGLT2 inhibitors, initially developed for managing hyperglycemia in type 2 diabetes patients, have recently been recognized for their benefits in CKD management (32,38). When combined with ACE/ARBs inhibitors, these agents have shown a capacity to reduce cardiovascular mortality and heart failure events (39,40). Additionally, they have demonstrated a potential in slowing the progression of kidney dysfunction in patients, irrespective of their diabetic status (41,42). Given the limited data on SGLT2 inhibitors cost-effectiveness in CKD management and limited incorporation into standard care practices, we adopted a conservative approach in assuming that the efficacy of SGLT2 inhibitors on kidney function is comparable between patients with and without diabetes. Our analysis specifically considers the introduction of dapagliflozin or analogous SGLT2 inhibitors to the conventional ACE/ARBs treatment regimen aligning with the intervention parameters outlined in related clinical trials (33,39,42). The efficacy of SGLT2 inhibitors on the rate of eGFR decline, all-cause mortality, and associated annual costs is provided in Table 1.”

(Page 18, Line 329 thorough 333, Results): “In our scenario analysis where SGLT2 inhibitors are added to the existing treatment regimen, annual universal screening beginning at age 30 became cost-saving compared with the status quo, despite the substantially higher annual medication costs of the new treatment regimen. When compared to the existing standard of care, the combined treatment strategy resulted in a reduction in total annual costs by approximately $516 per person. Moreover, this approach significantly enhanced health outcomes, decreasing the total DALYs and increasing the total QALYs by over 13% and 22%, respectively. It is important to note, however, that while these results demonstrate the promising role of SGLT2 inhibitors in the early management of CKD, their use in this context is still emerging. Current clinical trials and ongoing research are essential to further validate these findings and assess the full potential of SGLT2 inhibitors as a standard component in CKD treatment protocols.”

(Page 21, Line 387 thorough 395, Discussion): “CUSUMGFR-based screening was found to be cost-saving when new and relatively expensive therapy options such as SGLT2 inhibitors are introduced to the treatment regimen. This could be of importance in redefining the CKD management protocols and expanding the standard of care for CKD by adding more advanced, high-efficacy therapies despite higher treatment costs. However, it is important to acknowledge that SGLT2 inhibitors, while showing promise, are still in the early stages of being integrated into CKD management and additional clinical trials and research are needed to fully establish their role. Nonetheless, our study indicates that the inclusion of SGLT2 inhibitors in standard care could be beneficial for managing early stages of CKD, suggesting a potential shift towards more proactive and effective treatment strategies in the future.”

3. Representation of Diverse Populations: The study uses a nationally representative synthetic cohort. However, ensuring representation of diverse ethnicities and socioeconomic backgrounds could enhance the generalizability of the findings.

Response: Thank you for this important comment. We have now added more discussion regarding the limitation of the population simulated in this model and the cautions of generalization when interpreting the findings from our study in the limitation section: 

(Page 22, Line 406 thorough 409, Discussion): “Additionally , these datasets may not fully capture variations in awareness and comorbidities among underrepresented minority groups, potentially limiting our model's capacity to accurately reflect disparities and generalize the findings for an average population to diverse groups of different racial, ethnic, and socioeconomic backgrounds (52)”

4. Data Sources and Limitations: While the study utilizes NHANES data, discussing the limitations of these data sources, such as potential biases or gaps, would provide a clearer understanding of the study’s scope.

Response: Thank you for your comment. We added more discussion regarding the biases of self-reported NHANES data as a limitation to our study in the discussion section. 

(Page 22, Line 403 thorough 406, Discussion): “Third, our study parametrization relied on several data sources including the NHANES study, which despite being nationally representative, incorporates a significant proportion of self-reported information. Self-reported data may introduce biases, as they are subject to the accuracy of individual recall and personal perceptions (51)”

5. Long-term Projection Accuracy: The study projects outcomes over a lifetime horizon. Acknowledging and addressing uncertainties in long-term projections, like changes in healthcare practices or demographics, would be beneficial.

Response: Thank you for your comment. We expanded the discussion section to capture the inherent uncertainties in long-term projections such as reimbursement rates changes, clinical methods changes, etc. and noted that the study's findings should be interpreted with this consideration in mind.

(Page 22, Line 415 thorough 419, Discussion): “The dynamic nature of healthcare, with potential future changes in reimbursement rates, advancements in clinical methods, and evolving treatment strategies such as the emergence of SGLT2 inhibitors as new, effective drugs for CKD management, underscores the need for flexible and forward-thinking screening strategies. Moreover, there is a pressing need for practical guidelines that are fast, cost-effective, easy to interpret and implement, that do not place an additional burden on providers and healthcare systems.”

6. Comparison with Other Screening Methods: The study focuses on CUSUMGFR-based screening. Comparing its effectiveness and cost with other potential screening methods could provide a more comprehensive view of its relative benefits.

Response: Thank you for your comment. We agree that cross-comparing multiple screening modalities is a valuable analysis in general; however, in our specific problem setting of screening for chronic kidney disease, such comparisons may not be feasible, because screening in early stage of CKD and asymptomatic cases has not been widely practiced yet and most existing screening tests for urinary protein (micro- or macroalbuminuria) and serum creatinine (through eGFR) are primarily for patients already with mild kidney damage. In fact, introducing screening based on a simple blood test for early stage and asymptomatic population is the key and innovative idea of this study, to understand the value of such a simple screening test that can be conveniently built in clinical workflow of primary care. 

7. Policy Implications and Practicality: While the study suggests the cost-effectiveness of CUSUMGFR-based screening, discussing the practicality of implementing such policies in real-world healthcare settings would enhance its applicability.

Response: Thank you for this great suggestion. We have expanded our discussion on the implications of CUSUMGFR-based screening in real-world healthcare settings. 

(Page 22, Line 415 thorough 419, Discussion): “The dynamic nature of healthcare, with potential future changes in reimbursement rates, advancements in clinical methods, and evolving treatment strategies such as the emergence of SGLT2 inhibitors as new, effective drugs for CKD management, underscores the need for flexible and forward-thinking screening strategies. Moreover, there is a pressing need for practical guidelines that are fast, cost-effective, easy to interpret and implement, that do not place an additional burden on providers and healthcare systems.”

---

## [Editor Report · Decision Letter 1]

9 Feb 2024

Cost-effectiveness of Screening for Chronic Kidney Disease Using a Cumulative eGFR-based Statistic

PONE-D-23-27416R1

Dear Dr. Griffin,

We’re pleased to inform you that your manuscript has been judged scientifically suitable for publication and will be formally accepted for publication once it meets all outstanding technical requirements.

Kind regards,

Yuri Battaglia

Academic Editor

PLOS ONE
---

## [Editor Report · Acceptance letter]

29 Feb 2024

PONE-D-23-27416R1 

PLOS ONE

Dear Dr. Griffin, 

I'm pleased to inform you that your manuscript has been deemed suitable for publication in PLOS ONE. Congratulations! Your manuscript is now being handed over to our production team.

Kind regards, 

on behalf of

Prof. Yuri Battaglia 

Academic Editor

PLOS ONE